# A Ribosome Interaction Surface Sensitive to mRNA GCN Periodicity

**DOI:** 10.3390/biom10060849

**Published:** 2020-06-03

**Authors:** Kristen Scopino, Elliot Williams, Abdelrahman Elsayed, William A. Barr, Daniel Krizanc, Kelly M. Thayer, Michael P. Weir

**Affiliations:** 1Department of Biology, Wesleyan University, Middletown, CT 06459, USA; kscopino@wesleyan.edu (K.S.); elliot.james.williams@gmail.com (E.W.); aelsayed@wesleyan.edu (A.E.); wbarr@wesleyan.edu (W.A.B.); 2Department of Mathematics and Computer Science, Wesleyan University, Middletown, CT 06459, USA; dkrizanc@wesleyan.edu (D.K.); kthayer@wesleyan.edu (K.M.T.); 3College of Integrative Sciences, Wesleyan University, Middletown, CT 06459, USA; 4Department of Chemistry, Wesleyan University, Middletown, CT 06459, USA

**Keywords:** ribosome translocation, molecular dynamics, mRNA GCN periodicity, A-site decoding center, codon adjacency

## Abstract

A longstanding challenge is to understand how ribosomes parse mRNA open reading frames (ORFs). Significantly, GCN codons are over-represented in the initial codons of ORFs of prokaryote and eukaryote mRNAs. We describe a ribosome rRNA-protein surface that interacts with an mRNA GCN codon when next in line for the ribosome A-site. The interaction surface is comprised of the edges of two stacked rRNA bases: the Watson–Crick edge of 16S/18S rRNA C1054 and the adjacent Hoogsteen edge of A1196 (*Escherichia coli* 16S rRNA numbering). Also part of the interaction surface, the planar guanidinium group of a conserved Arginine (R146 of yeast ribosomal protein Rps3) is stacked adjacent to A1196. On its other side, the interaction surface is anchored to the ribosome A-site through base stacking of C1054 with the wobble anticodon base of the A-site tRNA. Using molecular dynamics simulations of a 495-residue subsystem of translocating ribosomes, we observed base pairing of C1054 to nucleotide G at position 1 of the next-in-line codon, consistent with previous cryo-EM observations, and hydrogen bonding of A1196 and R146 to C at position 2. Hydrogen bonding to both of these codon positions is significantly weakened when C at position 2 is changed to G, A or U. These sequence-sensitive mRNA-ribosome interactions at the C1054-A1196-R146 (CAR) surface potentially contribute to the GCN-mediated regulation of protein translation.

## 1. Introduction

Ratcheting movements and conformational changes of the ribosome large and small subunits are hypothesized to facilitate the threading of mRNA through the ribosome so that its codons can be translated into the protein product. In addition to the ratcheting rotation of the small subunit relative to the large subunit, swiveling of the head region relative to the main body/platform of the small subunit is thought to facilitate movement of the tRNA and mRNA from the Aminoacyl-tRNA (A) site (decoding center) to the Peptidyl-tRNA (P) site of the ribosome [1,2]. Dynamic interactions of the mRNA with ribosome surfaces are likely to be important for successful threading of the mRNA, which occurs during the translocation step accompanied by GTP hydrolysis of prokaryote EF-G-GTP or eukaryote eEF2-GTP [3,4]. Here, we present evidence that there is a ribosome interaction surface that is transiently tethered to the A-site tRNA anticodon, and that interacts with the mRNA +1 codon that is next in line to enter the A-site. This interaction suggests a framework for the potential modulation of protein translation rates.

Cryo-EM studies have suggested that the yeast ribosome has several intermediate stages of translocation ratcheting. Clustering analysis of very large numbers of cryo-EM images revealed five dominant intermediate stages (stages I through V; [5]). These intermediate structures are assumed to represent structures with local energy minima that the ribosome tends to sample during its progression through translocation. The ratcheting rotation angle decreases progressively from stages I to V (10°, 5°, 5°, 1°, 0.5°) paired with changes in the head swivel angle (12°, 17°, 17°, 14°, 1°) [5].

In this study, we assessed how the mRNA may interact with these intermediate structures using a subsystem of the ribosome (Figure 1A). We were particularly interested in how mRNA sequences might interact with the surface of the ribosome’s mRNA entrance or exit tunnels, because the codons immediately following the translation start codon, referred to as the ‘ramp’ region [6,7], have a tendency to have a three-nucleotide repeating pattern; GCN (where N represents A, C, G or U) [8]. A subtle GCN periodicity has been detected throughout ORFs [9,10], and we have found that the GCN periodicity is particularly pronounced in the ramp regions of genes exhibiting high protein expression [8]. These observations raise the hypothesis that mRNA codons might dynamically interact with the mRNA tunnel surface before or after passing through the decoding center A and P sites. These interactions, sensitive to mRNA sequence, could be particularly important in modulating translocation events, and hence rates of protein translation.

Consistent with this idea, Abeyrathne et al. [5] report in their cryo-EM characterization of the translocation intermediates that C1274 of 18S rRNA in *Saccharomyces cerevisiae* (henceforth referred to as C1054, the *Escherichia coli* 16S rRNA counterpart located in helix 34 in the small subunit head region) can base pair with G at the first nucleotide of the codon next-in-line to enter the A-site of the ribosome decoding center (+1 codon, Figure 2). While this interaction could help to explain the trend towards GCN periodicity in the ramp region, we were curious whether additional interactions in the mRNA entrance tunnel might also contribute to the prevalence of this striking periodicity.

Using molecular dynamics (MD) simulations and molecular modelling, we studied the neighborhood of the mRNA entrance tunnel that the mRNA encounters just before reaching the decoding center A-site. Our MD analysis has shown mRNA-rRNA and mRNA-ribosomal-protein interactions that are sensitive to the GCN periodicity in ORF ramps. Using mRNA with GCN codons following the A-site codon, we confirmed the base-pairing interaction between C1054 and G1 of the +1 codon next in line for the A-site. This interaction occurs in the early stages of protein synthesis translocation. We also found that the C nucleotide at the second position of the +1 codon has a specific Watson–Crick/Hoogsteen [14] base pairing with 18S rRNA A1427 (*Escherichia coli* 16S rRNA A1196, also in helix 34) which base stacks with C1054 and is highly conserved as A in 16S and 18S rRNA (or very occasionally U, as in *Thermus thermophilus*). Stacked on the other side of A1196 is the planar guanidinium group of R146 of yeast ribosomal protein S3 (Rps3, also in the small subunit head region, and conserved in eukaryotes) which also has a sequence-specific interaction with this C at position 2 of the +1 codon (Figure 2).

## 2. Materials and Methods

### 2.1. System for Study

In order to simulate the dynamics of the decoding center A-site region of the ribosome, we used the AMBER molecular modeling package [15,16,17] to create and test a subsystem of the ribosome. We considered a subsystem of the ribosomal structure (PDBID 5JUP) encompassing a 40 Å radius shell around the 16S/18S G530 nucleotide located in the mRNA tunnel immediately adjacent to the decoding center A-site. The distance constraint was used to select portions of the neighboring chains. This was further curated to include additional nucleotide and amino acid residues outside of the 40 Å shell that were between chain segments in order to reduce the overall number of artificial chain breaks in the subsystem (to 23 chains; Appendix A); additional residues were also added to ensure that chain breaks introduced by the procedure were located at a distance from the system of study—the decoding center neighborhood. Chain terminal residues were used to cap the chain breaks. Throughout the simulation procedure, 20 kcal/mol Å^2^ restraints were used on the outer residues (Figure 1, Appendix A), resulting in a fixed shell surrounding the native ribosomal environment for the decoding center and mRNA. The same set of residues were used for our simulations of the subsystems of translocation stages I through V (PDBID 5JUO, 5JUP, 5JUS, 5JUT and 5JUU).

The mRNA sequences of these cryo-EM structures have an AUG codon and a CCU codon in the P- and A-sites, respectively, followed by 5′GCU AAC. Using the t-LEaP application in the AMBER simulation package, the last codon was changed from AAC to GCC in all five structures (mRNA sequence 5′AUG CCU GCU GCC) because the available sequence did not match the GCN pattern. For the stage II subsystem (5JUP), we also made mRNA substitutions 5′AUG CCU GAU GCC, 5′AUG CCU GGU GCC, and 5′AUG CCU GUU GCC (see results). These nucleotide substitutions were made in t-LEaP by retaining common atom coordinates and growing in the rest of the residue (using standard template geometries from the parameter file ff14SB [18,19]) to preserve the residue’s relative orientation of the plane of the base with respect to the backbone. In addition, since His699 of elongation factor eEF2 is post-translationally modified to the non-standard residue diphthamide, the corresponding residue in our subsystem (His483) was parametrized using AMBER’s antechamber.

In order to resolve close atom contacts, a series of rounds of energy minimization were carried out. The optimized protocol entailed a ramp down of restraint weights and run lengths for a total of 11 rounds, each restrained to the last structure of the previous round (Appendix A). Specifically, 20,000 steps of minimization, beginning with 2500 steps of steepest descent, followed by the balance in conjugate gradient, were performed with a restraint weight of 100 kcal/mol Å^2^, followed by 10,000 steps of minimization (2500 steepest descent) with a weight of 75 kcal/mol Å^2^, 5000 steps (2500 steepest descent) at 65 kcal/mol Å^2^, 3000 steps (2500 steepest descent) at 55 and 45 kcal/mol Å^2^, 2000 steps (2000 steepest descent) at 30, 20, 15, 10, 5 and 1 kcal/mol Å^2^ restraints. The optimized protocol was empirically designed by monitoring the slope of the 5JUP subsystem’s total energy as a function of step, and convergence for a particular round was assumed when the slope asymptotically approached 0. Convergence on a number of rounds was assumed when adding more rounds resulted in little change from the previous round. Drops in the weight constraints (between rounds) were adjusted so that the total drop in energy was less than that of the previous round in the same or fewer number of steps. If this criterion was not met, an additional round with a smaller restraint drop was introduced. This minimization protocol was applied to all of the tested subsystems (different translocation stages and nucleotide substitutions). This energy minimization was followed by a 20 ps heating step, with all residues restrained at 20 kcal/mol Å^2^, and a time step of 2 femtoseconds, where velocities were assigned. Finally, the system was equilibrated for 3 ns with inherited velocities, with all residues restrained at 20 kcal/mol Å^2^, and a time step of 1 femtosecond. The resulting structure was used as the point of departure for MD, in which residues were unrestrained except for those in the onion shell. For translocation stage II, some MD runs terminated early, most likely due to high energy interactions typically observed at the interface between the unrestrained residues and the onion shell. We confirmed that runs could be stabilized without changing behavior by adding 10 or 100 pico-sec of MD before fixing atomic coordinates for the onion shell restraint mask.

### 2.2. Specifications of MD Simulations

The AMBER16.0 and 18.0 simulation packages and AMBERTOOLS14 suite [15,16,20] were used with the ff14SB force fields, which implement ff99bsc0_chiOL3 for RNA [18,19,21]. Simulations were run using the parallelized version of the code implemented on graphical processing units (GPUs) [17,22]. The structures were set up using t-LEaP. The system was solvated in a truncated octahedral box with a minimum of 12 Å between the system and the periodic boundaries (providing at least two layers of solvation or two solvation shells). The solvation model was TIP3P water [23]. Electro-neutrality was achieved with Na^+^ counter-ions. SHAKE was used to dampen high frequency hydrogen motions. Periodic boundary conditions were implemented, and long range electrostatic interactions were handled with particle mesh Ewald summation [24]. The Berendsen algorithm maintained the 300 K temperature [25].

### 2.3. Subsystem Data Collection

Snapshots were collected in a trajectory file at 100 frames per ns, and analyzed using AMBER cpptraj rms, hbond and distance functions with default settings (Appendix A) [16,26]. We analyzed the ‘out’ (time-course of frames) and ‘avgout’ (averages across all frames) outputs of the hbond function, which measures H-bonds with an acceptor-to-donor heavy atom distance less than 3.0 Å and an angle cutoff of 135°. Base pair hydrogen bonds were counted along the Watson–Crick and Hoogsteen edges [14]. Hydrogen bonds to Arg 146 of Rps3 were counted for the NE and NH1, NH2 and N proton donors. Base pair distances were computed using the centers of mass of the hydrogen donor and acceptor heavy atoms along the Watson–Crick or Hoogsteen edges. The distances to the closest edge of the guanidinium group of R146 were measured using the centers of mass of (R146) NE and NH1, and of NE and NH2 proton donors, and the smaller of the two distances for each frame was used. Base stacking distances were computed as the distances between the centers of mass of the heavy atoms of the base rings which gave readings between 3 and 4 Å, similar to stacked DNA bases, suggesting that the planes of the stacked bases were approximately parallel. RMSD measurements were made using the backbone atoms of amino acids and nucleotides.

## 3. Results

### 3.1. A Subsystem of the Ribosome

Ribosome translocation was divided into five stages [5] based on cryo-EM analysis of yeast ribosomes containing a viral internal ribosome entry site (IRES) RNA that mimics the mRNA and tRNA. For each of the five translocation intermediates (*Saccharomyces cerevisiae* PDB structures 5JUO, 5JUP, 5JUS, 5JUT and 5JUU), we set up a subsystem of 495 residues (183 nucleotides, 312 amino acids; Appendix A) centered around G530 of the highly conserved 530 loop, a landmark immediately adjacent to the decoding center A-site (Figure 1A,B). The subsystem contains 5.1% of the atoms of the ribosome.

Atomic coordinates from the cryo-EM structures were parameterized with t-LEaP, and any missing atoms were added from the standard templates. The structure and the local positioning of atoms was resolved through energy minimization (Appendix A). Before proceeding to MD analysis, we examined the static structures of the subsystem of the translocation intermediates using PyMol [11]. Highlighted regions of interest—the codon–anticodon, C1054 and A1196 of 16S/18S rRNA, and R146 of yeast Rps3—were examined across translocation stages I through V (Figure 1B illustrates stages I and V). The intermediate structures showed gradual translocation movement of the A-site codon–anticodon relative to the ribosome structure, and provided initial views of the transient adjacencies of C1054, A1196 and R146 with the +1 codon next in line to enter the A-site. The cryo-EM structures showed H-bonds between the first nucleotide of the +1 codon and C1054 in stages I and II, and showed an H-bond between this nucleotide and A1196 in stage I. These interactions were not observed in the later stage structures, and by stage V, C1054, A1196 and R146 were close to the +2 codon (not shown). These striking observations of ribosome–mRNA interactions in the static cryo-EM structures suggested that it would be valuable to examine the dynamic behaviors of this neighborhood in the five translocation stages.

Although our subsystem of the ribosome did not include possible long-range allosteric interactions, we hoped to represent in the subsystem some of the large-scale conformational changes in the different translocation stages. To ensure that our MD analysis would reflect the conformations of each stage, we applied an ‘onion shell’ restraining force of 20 kcal/mol Å^2^ to residues at the surface of the subsystem (Figure 1A, Appendix A), leaving the internal 173 residues unrestrained to demonstrate dynamics, and providing a buffer of at least one residue between the onion shell and C1054, A1196 and R146. The restrained residues are illustrated as spheres in Figure 1A. Examination of multiple MD runs for each of the five translocation stages showed that the onion shell was stable over time. RMSD measurements relative to the starting structure after heating to 300 K showed that the onion shell was stable, with RMSD values between 0.74 and 0.86 Å (compared to the reference structure) depending on the subsystem stage. For stages I through V, the unrestrained residues inside the onion shell stabilized within 10 to 15 ns to RMSD < 2 Å, suggesting that the systems had reached equilibrium (Appendix A). The stabilized structures of stages I-V after 15 ns of dynamics were assumed to represent the biological structures of the translocation intermediates, and were analyzed as described below.

### 3.2. C1054 Base Pairs to the mRNA

Since cryo-EM translocation structures [5] showed base pairing of C1054 to G at position 1 of the +1 codon next in line for the A-site (henceforth referred to as G1), we examined this interaction in our MD runs of stages I through V. Stable C1054:G1 base pairing was observed in the molecular dynamics of stages I and II, but not III, IV or V (Figure 3C and Figure 4A). After stage II, the C1054:G1 interaction becomes weak, as indicated by the low frequencies of frames showing H-bonding. Focusing on stages I and II, we observed pronounced hydrogen bonding in multiple MD runs starting with different randomly-assigned velocities. We compared the distributions of the numbers of hydrogen bonds over time for C1054:G1 and the middle nucleotide of the codon–anticodon base pairing, and found that the distributions were similar at the beginning of translocation (stages I and II; Figure 3A and Figure 4A), although occasional MD runs in translocation stage II showed little C1054:G1 base pairing (Appendix A). Integrating MD frames over time from multiple runs, we analyzed H-bonds using AMBER cpptraj and found that the average number of hydrogen bonds per frame was 1.9 (stage I) and 1.3 (stage II) for C1054:G1, compared to 2.2 (stage I) and 2.0 (stage II) H-bonds per frame for position 2 of the codon–anticodon C:G base pair (Figure 4A). Correspondingly, the average distances between the heavy atoms that form base-pair H-bonds was 2.9 Å (stage I) and 3.4 Å (stage II) for C1054:G1, compared to 3.0 Å (stages I and II) for the codon–anticodon position 2 base pair, reflecting the above time-integrated H-bond counts (Figure 4B; the distance between H-bonded heavy atoms < 3 Å, as reported by cpptraj). The C1054:G1 base pairing switched between two dominant behaviors, characterized by either 0, 1 or 2 H-bonds, or 0, 1, 2 or 3 H-bonds. At later translocation stages (III, IV and V) the distance between C1054 and G1 increased substantially (Figure 4B).

These observations suggest that, during the early stages of ribosome translocation (stages I and II), C1054 can base pair to position 1 of the next A-site codon when there is a G at this position which conforms to the canonical repeated GCN. The interaction is most pronounced at stage I, which correlates with the period when the ribosome shows the most pronounced ratcheting rotation (10°) of the small subunit relative to the large subunit [5].

### 3.3. A1196 and Rps3 R146 Hydrogen Bond to the C2 mRNA Nucleotide

Our molecular dynamics revealed that C at position 2 of the +1 codon next in line for the A-site (henceforth referred to as C2) base pairs with A1196 (A1427 in *Saccharomyces cerevisiae* 18S rRNA). The hydrogen bonding between C2 and A1196 occurs through a trans Watson–Crick/Hoogsteen base pair (‘reverse Hoogsteen’), where N6 of adenine donates a hydrogen to N3 of cytosine, and N4 of cytosine donates a hydrogen to N7 of adenine [14] (Figure 2).

This C2:A1196 base pairing was observed in translocation stages I through V, but was stronger in stages I, II and V (Figure 3D and Figure 4A). Correspondingly, the distance between the centers of mass of the A1196 Hoogsteen edge and C2 Watson–Crick edge (donor and acceptor heavy atoms) dropped to an average of 4.8 Å at stage II, and to near 3 Å for some runs (Figure 4B, Appendix A). The interaction was observed in some but not all of the independently-initiated molecular dynamics simulations in which velocities were randomly assigned to the starting structure (Appendix A).

Our MD runs also showed that C2 interacts with R146 of the yeast Rps3 (ribosomal protein S3), whose planar guanidinium R group is stacked immediately adjacent to the A1196 base. C2 forms frequent H-bonds with hydrogen-donor nitrogen atoms of the guanidinium R group (Figure 2, Appendix A). This interaction was observed in stages I and II (Figure 3E and Figure 4A). Indeed, the average distance between the donor nitrogen atoms of the guanidinium group and the C2 Watson–Crick edge heavy atoms was 4.7 Å in stage II and 6.3 Å in stage I (Figure 4B). The average distance was less than 4 Å in some of the individual MD runs at both stages (Appendix A), and frequently diminished to 3 Å permitting H-bonding.

Integrating the average number of H-bonds of A1196 or R146 with C2 over 1-ns intervals showed that, in some MD runs, the system switched multiple times between H-bonded and not H-bonded, suggesting a low energy barrier between these sub-state behaviors (Figure 4C, Appendix A).

### 3.4. C1054-A1196-R146 Stacking Defines an mRNA Interaction Surface

The pairing of G1 to C1054 and C2 to A1196 and R146 is facilitated by the base stacking of C1054 and A1196 and stacking of A1196 with the planar guanidinium group of R146 (Figure 2B and Figure 5). This stacking defines an interaction surface for the mRNA consisting of the Watson–Crick edge of C1054, the Hoogsteen edge of A1196, and an edge of the guanidinium group, which are adjacent and parallel to each other (Figure 2, henceforth referred to as the CAR—C1054-A1196-R146—interaction surface). The C1054-A1196 base stacking is very stable. In MD runs of translocation stages I and II, C1054 and A1196 are stacked most of the time, with very occasional loss of stacking (Figure 5, Appendix A). Correspondingly, the average distances between the centers of mass of the stacked bases is 3.9 Å (stage I) and 4.0 Å (stage II). C1054-A1196 stacking is less stable in stages III through V, and MD runs show switching between stacked and unstacked behaviors (Appendix A). C1054 and A1196 are on two separate bulges of loop 34 of the 16S/18S rRNA that are brought together and positioned through the C1054-A1196 base stacking interaction. The stacking positioning of C1054 and A1196 is aided by the frequent H-bonding of A1196 O2′ with OP2 of the C1054 phosphate (Figure 5C, Appendix A). The stacking of Rps3 R146 with A1196 is also extremely stable through most of translocation (Figure 5B, Appendix A), providing stability to the mRNA interaction surface. R146 is stacked adjacent and close to A1196. Indeed, the average distance between the centers of mass of the A1196 base and the guanidinium group of R146 was 4.2 Å (stage II), and a little higher at stages I, III and IV. The center-of-mass distances between A1196 and R146 are considerably higher at stage V (average of 8 Å; Figure 5A) when A1196, but not R146, shows H-bonding to C2 (Figure 3 and Figure 4). At translocation stage V, the swiveling of the head region relative to the main body of the ribosome small subunit is at a minimum (1°) [5]. A1196 and R146 are likely influenced by the swivel angle, since they are located in the head region, close to the small subunit main body/platform.

### 3.5. The A-Site tRNA Anchors the C1054-A1196-R146 Interaction Surface

At the beginning of translocation during stages I and II, the stacked CAR interaction surface is positioned like an extension of the A-site. Indeed, C1054 shows base stacking with the adjacent wobble position anticodon nucleotide of the tRNA (Figure 2). Stacking of C1054 with the wobble anticodon nucleotide is quite stable, with an average center-of-mass difference between the two bases of 5.4 Å (stage I) and 4.6 Å (stage II) (Figure 5A,B). This distance hovers just below 4 Å when the bases are stacked, but occasionally switches to greater than 5 Å when C1054 no longer stacks with the wobble anticodon base (Figure 5B, Appendix A). Anchoring is also aided by H-bond interactions between the wobble anticodon nucleotide and C1054 (see Figure 5C and legend). After translocation stage II, the CAR interaction surface remains intact (pi stacked), but is no longer anchored to the A-site anticodon wobble nucleotide, reaching a separation distance of 18 Å by stage V (Figure 5, Appendix A). The loss of anchoring after stage II correlates with the reduction in ratcheting rotation angle between the large and small ribosome subunits [5].

Anchoring of the C1054-A1196-R146 interaction surface to the tRNA anticodon helps align the interaction surface with the +1 mRNA codon next in line for the A-site. Indeed, C1054 stays very well base paired to G1, and the dominant H-bond interactions of A1196 and R146 are with C2 (Figure 6A).

### 3.6. Sequence Sensitivity of the C1054-A1196-R146 mRNA Interaction

The mRNA interactions of A1196 and Rps3 R146 were observed with C at position 2 of the +1 mRNA codon next in line for the A-site. This matches the GCN periodicity that is particularly pronounced in the initial codons of the ORFs of genes with high protein expression [8]. Since there is a strong preference for C and against G at position 2 of the codons, we were interested to examine how changing the nucleotide at position 2 might affect mRNA interactions with the CAR (C1054-A1196-R146) interface. To address this, we performed an MD analysis of translocation stage II subsystems in which the C at position 2 was changed to G, A or U. Stage II was used because the combined H-bonding of C2 to A1196 and R146 was highest at this stage (Figure 4A).

As illustrated in Figure 6, the Hoogsteen edge of A1196 can potentially base pair with G, A or U. However, compared to the A1196:C2 base pair, these base-pairing interactions require changes in the distances between the sugar-phosphate backbones (of A1196 and the mRNA), and the registration of the interacting base edges. A1196 base pairing to G also requires a 180-degree swivel of the Watson–Crick edge about the glycosidic bond (not shown). The edge of the guanidinium group of R146 presents two hydrogen-donor nitrogen atoms (NE and NH1 or NH2) which were observed to H-bond with the N3 and O2 hydrogen acceptors of C2. The Watson–Crick edges of A, G or U only offer one hydrogen acceptor, suggesting that R146 might not interact as effectively with these bases (Figure 6). Given the structural integrity of the CAR interaction surface conferred by the stacking of C1054, A1196 and R146, we suspected that changes in position 2 of the +1 codon next in line for the A-site might influence mRNA interactions with the surface at both positions 1 and 2 of the +1 codon.

After replacing C2 with G, A or U (G2, A2 or U2) in the mRNA, we carried out multiple MD runs (9 to 11 runs), starting with independent heating steps, the assignment of velocities to atoms, and equilibration. We monitored the behavior of the CAR interaction surface after equilibrating the systems to allow the replaced position 2 nucleotide to adjust in the system; the equilibration consisted of 3 ns of dynamics with all residue atoms restrained, and 15 ns of dynamics in which residues inside the restrained onion shell were unrestrained. RMSD profiles of the MD runs (for backbone atoms of the unrestrained residues) showed RMSDs below 2 Å, which became stable during these 15 ns (Appendix A). We examined the RMSD profiles of residues in the immediate vicinity of the substituted nucleotides (residues within 10 Å of A1196), and found that these RMSD profiles similarly stabilized within the 15 ns (Appendix A).

Compared to C at position 2 of the +1 codon next in line for the A-site (C2), replacement of this nucleotide with G2, A2 or U2 led to pronounced reductions in hydrogen bonding to A1196 and R146 (Figure 6, Appendix A). The reductions were particularly pronounced for the bulkier purine residues G2 and A2. Moreover, replacing C at position 2 led to slight reductions in the hydrogen bonding of G at position 1 of the +1 codon to C1054, although only the U2 reduction was significant (*t*-test *p* < 0.05; Figure 6). Interestingly, with the G2 substitution, G1 showed somewhat elevated H-bonding by heavy atoms away from the Watson–Crick edge, including O2′ interaction with NH1 of R146 (Figure 6B, Appendix A), reflecting an altered configuration of the CAR interface relative to the mRNA. Many of the MD runs with G2 or A2 substitutions showed stacking of the G2 (or A2) purine base with R146, creating an extended arch of stacking through to the A-site anticodon (Figure 6D, Appendix A), a configuration in which the CAR interface is rotated approximately perpendicular to the mRNA. In summary, replacement of position 2 in the +1 codon, an alteration of the canonical GCN pattern, has broad effects on the CAR interface, changing both its hydrogen bonding and stacking properties.

## 4. Discussion

### 4.1. The C1054-A1196-R146 (CAR) Interaction Surface

The present study suggests that, during ribosome translocation, the +1 mRNA codon which is about to enter the A-site has transient, sequence-sensitive H-bond interactions with a surface consisting of the edges of stacked rRNA residues 16S/18S C1054 and A1196 (*Escherichia coli* coordinates), as well as R146 of yeast S3 ribosomal protein Rps3. This C1054-A1196-R146 (CAR) interaction with the mRNA is strongest when the +1 codon conforms to GCN; replacement of the second nucleotide with G, A or U significantly weakens the interaction (Figure 6).

C1054 and A1196 are highly conserved in eukaryotes and prokaryotes (Appendix A), suggesting that their functions may be conserved. Less commonly, the ribosome has U at 1196, for example as in *Thermus thermophilus*. R146 (yeast coordinates) is highly conserved across eukaryotes (Appendix A), again suggesting a shared function. However, R146 conservation does not extend to prokaryotes, so it is not yet known whether they have an interaction surface equivalent to CAR.

Previous studies have suggested functional roles for C1054 and A1196 residues in protein translation in prokaryote and eukaryote systems. Cryo-EM analysis of translocation intermediates in yeast [5] showed specific base pairing of C1054 with G at the first position of the +1 codon about to enter the A-site. In early cross-linking experiments with bacterial ribosomes [27,28,29], C1054 and A1196 were found to cross link with mRNA just downstream of the A-site. Various mutants of the bacterial elongation factor, EF-G, block translocation with concomitant loss of protection of C1054 and A1196 [30]. Moreover, C1054 mutants have altered translation accuracies that depend upon codon contexts, as well as translation termination phenotypes [31]. Similarly, mutational analysis has shown that mutants of the yeast counterpart of C1054 (C1274) have translation fidelity phenotypes [32].

Mutations that replace R146 in yeast Rps3 have lethal or slow growth phenotypes (depending on the amino acid substitution), and can affect translation fidelity as well as translation initiation [33,34]. Mass spectrometry analysis has shown that R146 in yeast Rps3 can be methylated by Sfm1 methyl transferase [35], and this study also noted the stacking of R146 with A1196 and C1054. Similarly, the conserved R146 in human ribosomal protein S3 can be mono- or di-methylated [36]. It is possible that the methylation of the guanidinium group of R146 may disturb its stacking with A1196 and its H-bonding with the +1 codon.

### 4.2. Wobble Anticodon Nucleotide Anchors the C1054-A1196-R146 Interaction Surface

Stacking interactions between the C1054 and A1196 bases, and between the A1196 base and guanidinium group of R146 [37,38,39], provide geometric rigidity to the interaction surface. A stacking interaction between C1054 and the wobble anticodon nucleotide serves as an anchor for the CAR interaction surface, thereby aligning it adjacent to the anticodon so that it is poised to pair with the +1 codon about to enter the A-site. Augmenting the stacking interaction, C1054 can also H-bond with the wobble anticodon nucleotide (for example, the N2 atom of the wobble anticodon G often H-bonds with C1054 OP2; Figure 5C). The strength and consistency over time of these anchoring stacking and H-bond interactions will be crucial for the strength and sequence sensitivity of the CAR interaction with the mRNA. Since the CAR interactions depend on both the wobble anticodon nucleotide and mRNA +1 codon, this implies a dependency between adjacent codons, which is consistent with previous observations that translation efficiency can be a property of adjacent codons [40,41,42].

### 4.3. Regulation of the mRNA–CAR Interaction

We suspect that the strength of the mRNA–CAR interaction is under tight regulation, possibly through methylation of R146 [35] or modifications of the anticodon wobble nucleotide [43,44] which can be differentially regulated under different cellular conditions such as stress [45,46,47]. Methylation of R146 may reduce its stacking with A1196 resulting in reduced mRNA–CAR H-bonding. Modifications of the anticodon wobble nucleotide may alter CAR anchoring and mRNA–CAR H-bonding.

Interactions of the mRNA with the CAR surface during translocation are expected to affect the overall efficiency of ribosome translocation. However, whether the effects are positive or negative awaits future analysis. One could imagine that, during the ratcheting of the ribosome during translocation, strengthening the interaction with the CAR surface might improve the efficiency of mRNA threading through the ribosome. Alternatively, increasing H-bonding to the CAR surface could make ribosome-mRNA interaction too ‘sticky’, possibly hindering the threading of the mRNA and reducing translocation efficiency. We hypothesize that both scenarios are true, depending on the strength of the mRNA–CAR interactions and translation level of the gene [8]. For low-expression genes, which tend to have low or modest GCN periodicity in their ramp, increasing mRNA–CAR interactions may promote translation, whereas for high-expression genes, which tend to have strong GCN periodicity, increased mRNA–CAR interactions may depress translation. In this model [8], increased mRNA–CAR interactions under stress conditions would result in up- or down-regulation of translation, depending on a gene’s expression level and the strength of its ramp GCN periodicity. Since genes with high protein expression tend to have strong GCN periodicity in their ramp regions, translation of these genes would be depressed under stress conditions. The GCN periodicity is also pronounced immediately after start codons used for non-standard translation products of genes, including miniproteins initiated downstream of the normal start codon [48,49], suggesting that CAR-GCN interactions may contribute to the regulation of their expression.

## 5. Conclusions

In summary, our MD analysis of 495-residue subsystems of several stages of translocating ribosomes has uncovered an interaction surface that behaves like an extension of the A-site anticodon and interacts with the +1 codon about to enter the A-site. The CAR interaction surface consists of stacked 16S/18S residues C1054 and A1196, and Rps3 residue R146, and preferentially interacts with +1 codons with a GCN sequence. When C at position 2 of the codon is replaced with G, A or U, the mRNA–CAR interaction is much weaker, suggesting that the mRNA–CAR interaction may play a role in modulating the translation of GCN-rich mRNA sequences.

## Figures and Tables

**Figure 1 biomolecules-10-00849-f001:**
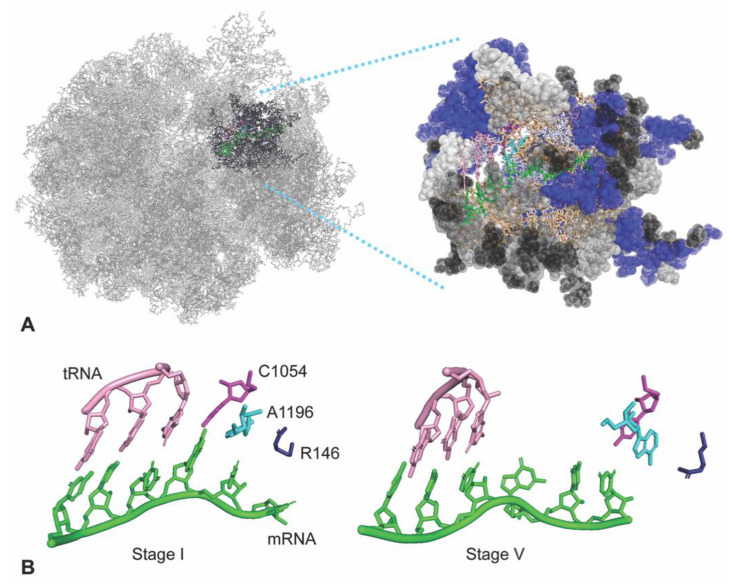
A ribosome subsystem used for molecular dynamics analysis of translocation stages. (**A**) A subsystem of 495 residues was defined for MD simulations for each translocation stage defined in cryo-EM analysis [5]; stage II cryo-EM structure 5JUP is illustrated here. Residues in an ‘onion shell’ are restrained with a force of 20 kcal/mol Å^2^ (spheres). Residues were visualized with PyMol [11] and color coded: restrained protein (blue), restrained nucleotides (grey), chain ends (black), unrestrained nucleotides (tan), unrestrained protein (light blue), mRNA (green), IRES mimicking A-site tRNA (pink). (**B**) Threading of the mRNA (green) through the decoding center is illustrated in translocation stage I compared to stage V (cryo-EM structures 5JUO and 5JUU). The mRNA and tRNA shift relative to 16S/18S rRNA highlighted residues C1054 (purple), A1196 (cyan) and Rps3 R146 (dark blue).

**Figure 2 biomolecules-10-00849-f002:**
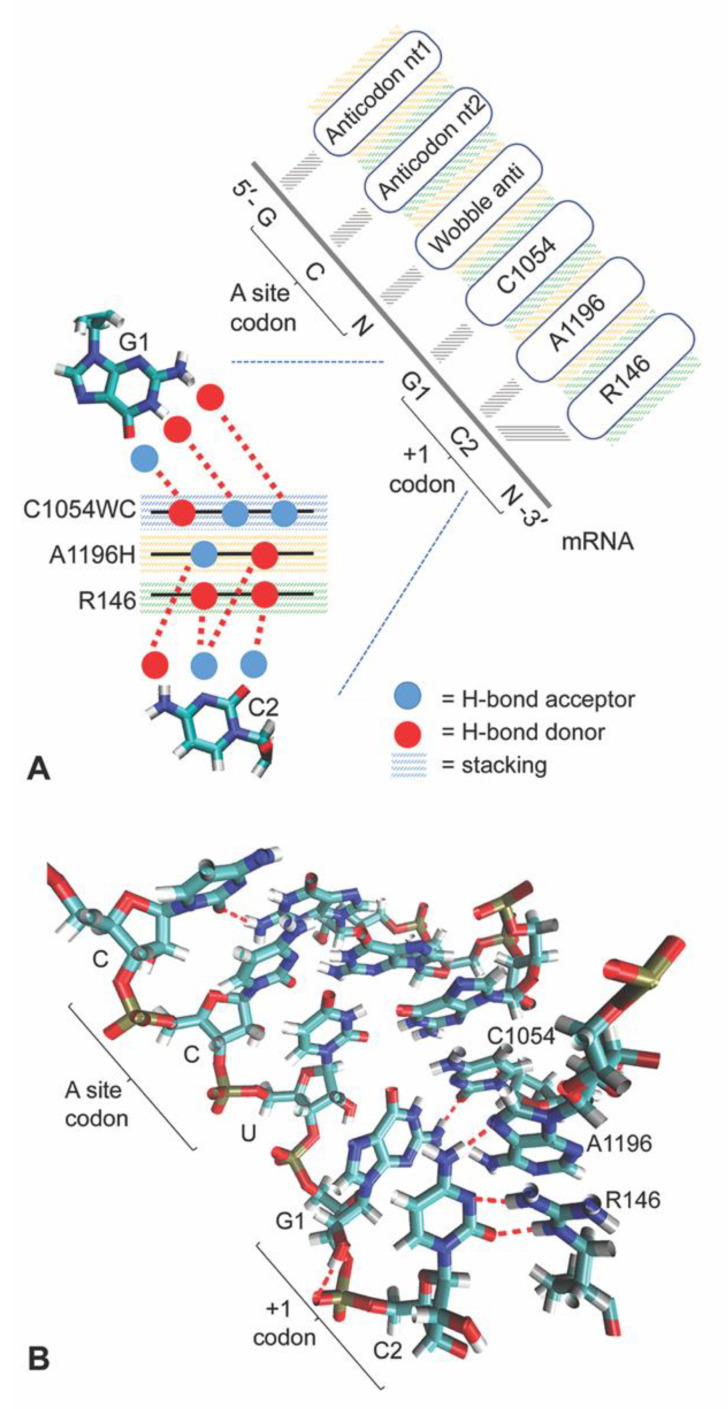
Molecular dynamics reveals the CAR interface, an interaction surface potentially involved in regulating translocation efficiency. (**A**) Model for H-bond interactions with G1 and C2 of the +1 codon, illustrating stacking interactions and H-bond donor and acceptor interactions at the CAR interface. C1054 interacts through its Watson–Crick edge (C1054WC), A1196 through its Hoogsteen edge (A1196H) and R146 through its guanidinium group. (**B**) Example of an MD frame illustrating H-bonding and stacking of C1054, A1196 and R146, and anchoring of this CAR surface through stacking with the wobble anticodon nucleotide. Molecular visualization was carried out with visual molecular dynamics (VMD) [12,13].

**Figure 3 biomolecules-10-00849-f003:**
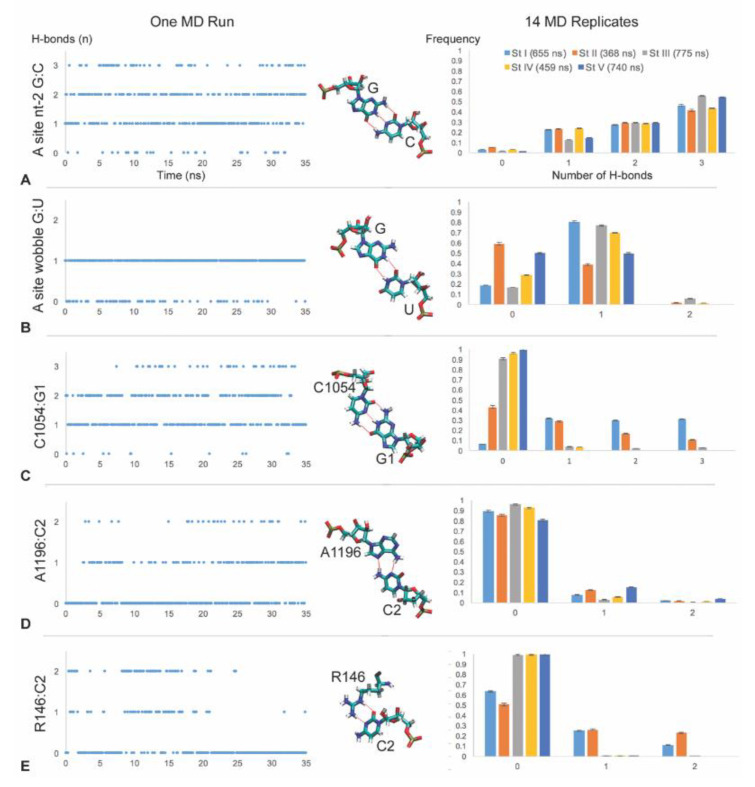
Dynamic H-bond interactions of C1054, A1196 and R146 with G1 and C2 of the +1 codon next in line to enter the ribosome A-site. H-bonding by the second (G:C) and wobble (G:U) nucleotides of the A-site are shown for comparison. The left-hand column shows profiles from one 35-ns MD run from translocation stage II, sampled at 10 frames per ns. The middle column shows the H-bond interactions (red dashes) from the same experiment. C1054 displays standard C:G base pairing between Watson–Crick edges. The Hoogsteen edge of A1196 base pairs to the Watson–Crick edge of C2. Proton-donating nitrogen atoms of the guanidinium group of R146 also hydrogen bond to C2. The right-hand column shows quantitation of H-bonds across all our MD experiments for translocation stages I through V. Each row (**A**–**E**) across the three columns illustrates the same H-bond interaction. Charts show the means of multiple MD experiments (see Appendix A), and error bars in this and the following figures show standard error. H-bonding by C1054 and R146 is most prominent in stages I and II, whereas A1196 shows interactions at all stages.

**Figure 4 biomolecules-10-00849-f004:**
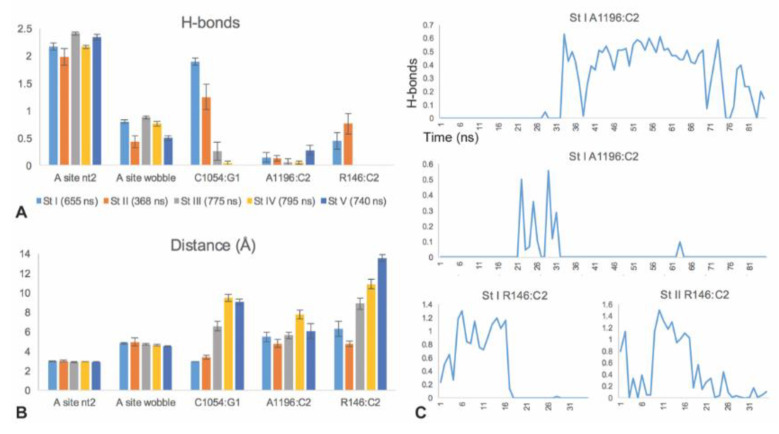
Stage specificity of CAR interactions. Multiple independent MD simulations were performed for each translocation stage. (**A**) Average numbers of H-bonds are shown for stages I–V. (**B**) Corresponding distances are shown between the heavy atoms participating in H-bonds. Distances are between centers of mass of Watson–Crick or Hoogsteen edges. For R146, we used the smaller of the two distances to the centers of mass of NH1,NE and NH2,NE in each frame. (**C**) Examples of MD runs exhibiting multiple transitions between H-bonded and not H-bonded, suggesting low energy barriers between the two sub-states. Frames were pooled in 1-ns bins.

**Figure 5 biomolecules-10-00849-f005:**
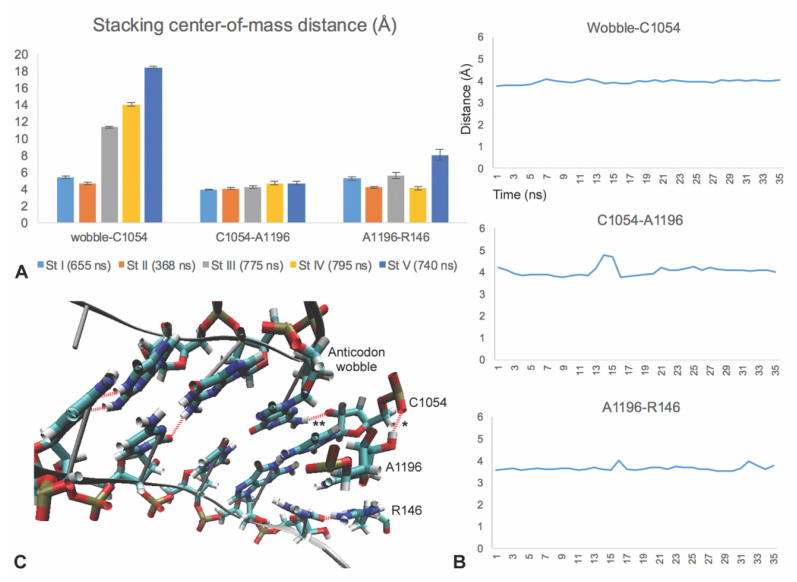
Stacking interactions. (**A**) Stacking between bases of C1054 and A1196 and the wobble anticodon base lead to short distances of about 4 Å between the centers of mass of the bases. Similarly, A1196 pi stacks with the planar guanidinium group of R146. The CAR interface is stacked to the wobble base in stages I and II, but loses stacking and moves away in stages III to V. (**B**) Examples of stage II profiles showing that a stacking separation distance of 4 Å is fairly stable over time. (**C**) H-bond interactions help stabilize and anchor the CAR interface. O2′ of the A1196 sugar often H-bonds to the OP2 of the C1054 phosphate (*; stage II structure viewed from above the CAR interface) (also see Appendix A). N2 of G in the anticodon wobble position often H-bonds to O2′ of the C1054 sugar (**). The latter interaction likely depends on the identity of the anticodon wobble nucleotide.

**Figure 6 biomolecules-10-00849-f006:**
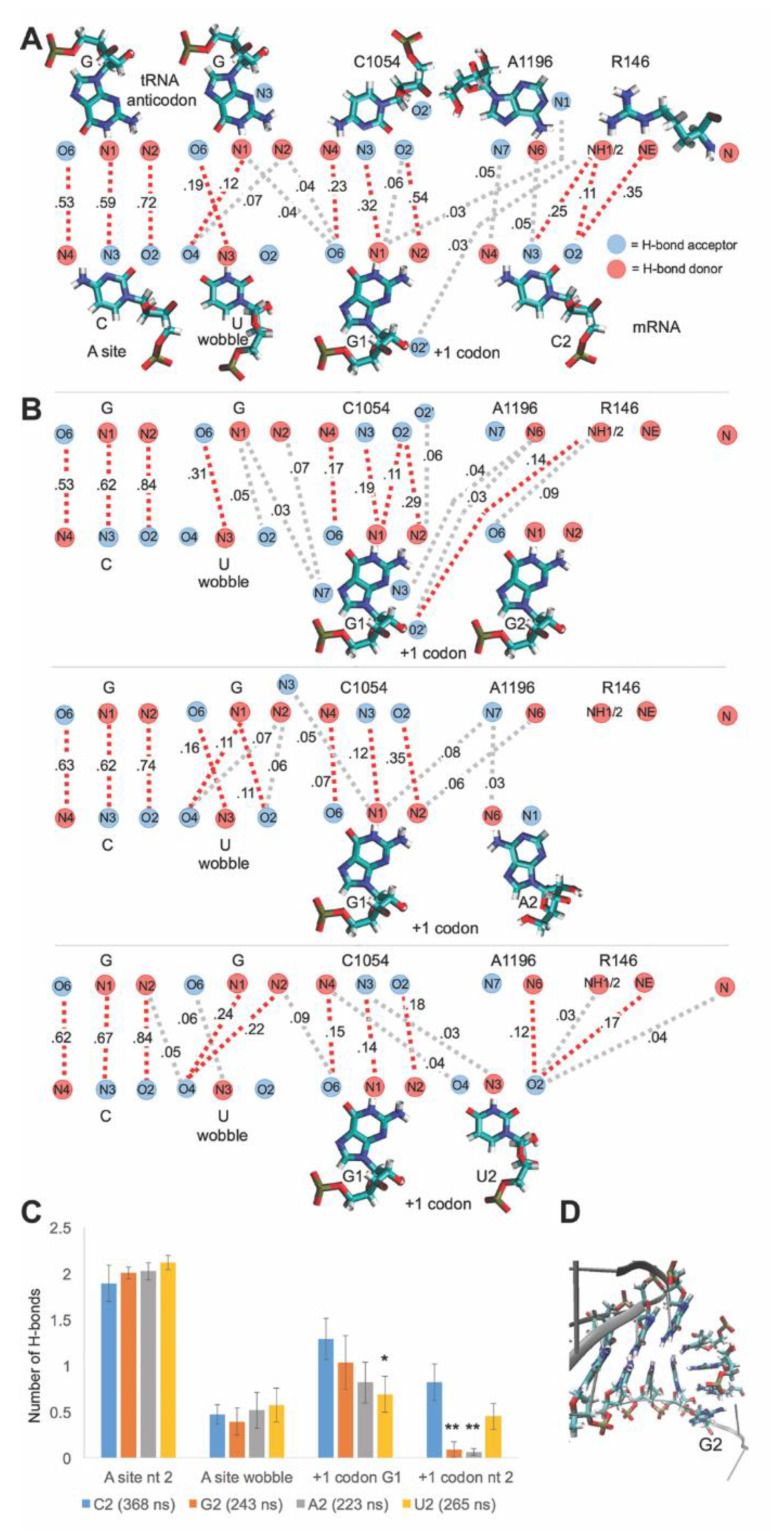
Sequence sensitivity of CAR interactions. (**A**) Frequencies of H-bond interactions with the +1 codon GCN during Stage II. Average frequencies are illustrated for 14 independent MD runs (see Appendix A) with red (frequency ≥ 0.10) or grey (0.025 < frequency < 0.10) dashes. The dominant interactions were G1 with C1054 and C2 with A1196 or R146, suggesting the good alignment of these residues. G1 also showed minor interactions with A1196 and the wobble anticodon G nucleotide. (**B**) C at position 2 of the +1 codon (C2) was replaced with G, A or U (G2, A2, U2) in the stage II structure. Multiple independent MD runs (9, 11 and 11 runs, respectively) were performed for each substitution. MD runs showed reduced H-bonding to the CAR interface with G2, A2 and U2 compared to C2. With G2, the adjacent G1 gained additional H-bond interactions (from N2, N3 and O2′) not involving its Watson–Crick edge. (**C**) Average numbers of H-bonds were graphed for each mRNA residue, including contributions from the aligned partner and the −1 and +1 cross registrations (e.g., f_G1:G_ + f_G1:C1054_ + f_G1:A1196_). Compared to C2, H-bonding to G2, A2 and U2 was reduced. Reductions were particularly pronounced for substitutions of the larger purine bases (G2 and A2; *t*-test: *p* < 0.01 **). Although the adjacent G1 also showed some reductions in H-bonding with G2, A2 and U2, the reductions were only significant for U2 (*t*-test: *p* < 0.05). Hence, the most pronounced H-bonding to the CAR interaction surface was observed with C2, which conforms to GCN at the +1 codon. (**D**) Example of the stacking of G2 with R146. Stacking extends from G2 through the CAR interface and A-site anticodon. Video 2 shows this MD run, during which the anticodon second and wobble nucleotides are transiently unstacked.

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
