# Peer review of "A Ribosome Interaction Surface Sensitive to mRNA GCN Periodicity"

_biomolecules, 2020, doi:10.3390/biom10060849_

Round 1
Reviewer 1 Report
The process of recognition of mRNA by the ribosome-tRNA complex is clearly dynamic. Therefore, it is not surprising that it has attracted the attention of researchers applying molecular dynamics modeling. In the manuscript by K. Scopino et al. "A Ribosome Interaction Surface Sensitive to mRNA GCN Periodicity" the authors present results of molecular dynamics simulations of the segment of the small subunit of yeast ribosomes associated with mRNA a specially modeled on the basis of IRES. It is important that simultaneously the authors published an experimental article in bioRxiv, in which they showed that the codon GCN is quite common among initial codons in ORF of different mRNAs. Using modern versions of MD simulation approaches the authors described interactions of this codon "when next-in-line for the ribosome A site" with the corresponding residues of the small ribosomal subunit. It is important that they were able to demonstrate the differences of these interactions when the ribosome transits from one intermediate stage of translocation process to another. The data are convincing and the conclusions are warranted by them. Thus the manuscript is worthy of publication.
I would like to make two suggestions concerning the reviewing manuscript.
1. I agree that in order to solve this problem, the authors did not need to work with the complete ribosome, as is necessary when you are analysing the entire translocation process (see, for example Bock et al, 2013. Nat.Struct.Mol.Biol., 20,1390-1396). The choice of a 40-Angstrom neighborhood from 495 residues is more or less adequate to the task, since it is unlikely that increasing the mobility of the neighborhood would break the nucleotide pairs or interaction with arginine. However, it still makes sense to perform preliminary optimization procedures for the entire ribosome. It is also confusing that the mobile core of this system is really quite small, 20 percent of the cut fragment. In other words, it is reasonable to ask the authors to justify the choice of this size, as well as to ask how many unrestricted residues in motion surrounded the interacting pairs of codon and ribosome. I.e., what is the thickness of the layer of the residues surrounding the studied residues, not counting positionally limited ones. Based on what I've looked at PyMOL, it's hardly more than two or three.
2. "Electro-neutrality was achieved with Na+ counter ions".
It should be done with K+ ions. The ribosome does not work when potassium ions are replaced with sodium ions. This is also typical for other ribozymes. I don't know if there are key potassium ions that coordinate with ribosome decoding center (as it does in its peptidyl transferase center). And I don't think the authors need to repeat all the calculations by replacing NaCl with KCl. But they need to keep this important fact in mind in the future. It should be also noted that the problems with KCL in MD simulations were solved with Grubmueller's group many years ago.
Reviewer 2 Report
In this manuscript, authors have proposed that residues on ribosomal surface (both nucleotide and amino acids) form a surface sensitive to recognizing GCN codons in mRNA open reading frames. This study was conducted completely in silico using techniques in molecular dynamics simulation. Using MD simulations, authors show that the mRNA interaction surface of the ribosome were composed of the Watson-Crick and Hoogsteen edges of bases C1054, and A1196 respectively of 16S rRNA and the guanidinium group of a conserved R146 of ribosomal small sub-unit protein Rps3. Authors here have constructed ribosome subsystems comprising a 40Å shell around the base G530 and MD simulations were performed on these sub-systems defined from previously described five intermediate stages of mRNA translocation during translation. Their results suggest that C1054 of 16S/18S interacts with the guanine in the +1 GCN codon; A1196 and R146 interact with the cytosine, while making base stacking interactions with each other and with the wobble base of the A-site tRNA anticodon stem loop.
This manuscript is very clearly written and the experimental procedures are adequately explained. Authors have also presented a detailed discussion of experimental results and interpretations. Better clarification is required is some parts of the manuscript. Some of my concerns about this work are explained below:
Major Points:
- In this study authors have used ribosome sub-systems made by isolating a shell containing the CAR element. By using only a sub-system, authors have excluded effects of distant features of the ribosome that can affect the CAR-mRNA interactions allosterically.
- Authors have shown that when there is a G, A or U in the +2 position of the upcoming GCN codon, its interaction with the CAR interaction surface is dramatically weakened. However, authors have not compared the GCN codon-CAR surface interactions to mRNAs with A, U or C in the +1 position.
Minor Points:
- Authors should edit the last paragraph on Page6 (lines 224-235) for more clarity bout stages. what is the significance of identifying number of H-bonds. What is the H-bonding values for a non-G nucleotide at the +1 position?
- Line 301: How do you explain the increased distance between A1196 and R146 at stage 5 in translocation.
- Line 342: the sugar-phosphate backbone of which residues.
